# Caco-2 Cell Response Induced by Peptides Released after Digestion of Heat-Treated Egg White Proteins

**DOI:** 10.3390/foods11223566

**Published:** 2022-11-09

**Authors:** Sara Benedé, Leticia Pérez-Rodríguez, Elena Molina

**Affiliations:** 1Instituto de Investigación en Ciencias de la Alimentación (CIAL, CSIC-UAM), Calle Nicolás Cabrera 9, 28049 Madrid, Spain; 2Departamento de Inmunología, Oftalmología y Otorrinolaringología, Facultad de Medicina, Universidad Complutense de Madrid, 28040 Madrid, Spain

**Keywords:** ovalbumin, ovomucoid, lysozyme, Caco-2 cells, heat treatment, digestion

## Abstract

The heat treatment of food proteins induces structural modifications that influence their interaction with human fluids and cells. We aimed to evaluate the Caco-2 cell response induced by peptides produced after digestion of heat-treated egg white proteins. In vitro digestion of ovalbumin (OVA), ovomucoid (OM), and lysozyme (LYS), untreated or previously heated, was performed. The digestibility of proteins and the response of Caco-2 cells exposed to peptides (<10 kDa) generated during digestion were evaluated. Intact OVA and LYS persisted after the digestion of native proteins, whereas OM was completely hydrolysed. A heat treatment at 65 °C for 30 min did not alter the digestibility of OVA, whereas at 90 °C for 3 min, protein degradation was favoured. The digestibility of OM and LYS was not affected by heat treatment. Peptides derived from OVA and OM digestion induced IL-6 and IL-8 production. OVA and LYS digestion promoted the expression of *Tslp*, and *Il6* and *Il33*, respectively. A heat treatment prior to OVA digestion reduced IL-6 production and Tslp expression. It was concluded that heat treatments can reduce the release of OVA-derived peptides, but not OM and LYS, with proinflammatory activity during digestion.

## 1. Introduction

Hen eggs are one of the commonest causes of food allergies in infancy and childhood among the European population [1] and, specifically, affects 0.9% of all children and 1.3% of children < 5 years old in the United States [2], having been found to be the most common food trigger of anaphylaxis in infants < 12 months of age [3]. There is currently no cure for an egg allergy, and the only way to prevent egg allergy symptoms is to avoid the consumption of eggs. However, strict avoidance of egg intake is difficult because, due to its excellent technological properties, it is used as an ingredient or additive in a large number of processed foods [4].

Ovalbumin (OVA), ovomucoid (OM), and lysozyme (LYS) have been identified as major allergens in hen eggs. OVA (Gal d 2) contains 385 amino acids and comprises 54% of the total proteins in egg whites. OM (Gal d 1) is a highly glycosylated molecule containing 186 amino acid residues and is considered the immunodominant allergen [5]. LYS (Gal d 4) is a glycosidase that contains 129 amino acids and is commonly used as a food and drug preservative due to its antibacterial properties [6]. 

The allergenic potential of egg white proteins may be affected by heat treatment, one of the most important food preparation methods that is usually applied to food to maintain the microbiological quality. The allergenic epitopes can be hidden, destroyed, or masked through conformational changes [7]. In addition, heat treatments can modify egg functionality and quality through structural transformations of egg proteins, which can also affect their digestibility within the gastrointestinal tract [8] and, consequently, their ability to sensitise and elicit an immune response [9].

The gastrointestinal epithelium interacts with food proteins after modification during digestion, leading, mainly, to peptides and amino acids, which are absorbed depending on size, polarity, and shape following four different routes: paracellular diffusion, transcellular passive diffusion, transcytosis, and carrier-mediated transport [10]. In addition, intestinal epithelial cells, which constitute the interface between the gastrointestinal content and the gut-associated lymphoid tissue, most likely determine the immune response to food [11]. Intestinal epithelial cells are able to secrete cytokines (e.g., IL-1, IL-18, IL-25, IL-33, and thymic stromal lymphopoietin—TSLP-) that are essential for the initiation of allergen sensitisation and promote the activation of DCs, type 2 innate lymphoid cells (ILC2), basophils, eosinophils, and mast cells, skewing the intestinal immune system toward a Th2 response [12]. 

Almost all studies evaluating the resistance of food allergens to digestion assess the survival of intact proteins, but very few studies examine the interaction between peptides and intestinal epithelial cells [9,13]. Considering that the heat treatment of egg white proteins induces structural modifications, which greatly influences their interaction with human fluids and cells, we aimed to assess the impact of this treatment on the digestibility of egg white proteins and the subsequent intestinal response elicited by the generated peptides.

## 2. Materials and Methods

### 2.1. Heat Treatment of Egg White Proteins

OVA, OM, and LYS (Sigma-Aldrich, St. Louis, MO, USA) were dissolved in Milli-Q water at 5 mg/mL with a pH of 7.0. Protein solutions were treated in a water bath at 65 °C for 30 min and at 90 °C for 3 min, mimicking the most common egg cooking conditions. After the heat treatments, the pH was readjusted to 7.0 and the solutions were freeze-dried until used (the experimental design of the study is shown in Figure 1).

### 2.2. In Vitro Gastrointestinal Digestion

Simulated gastrointestinal digestion was performed as described by Villas-Boas et al. [15]. The digested samples were fractionated using 10 kDa cut-off ultrafiltration units (Amicon Ultra, Millipore, Eschborn, Germany). The protein concentration of the fractions, analysed via the Kjeldahl method, was used to standardise their concentration for in vitro experiments.

### 2.3. SDS-PAGE Analyses

SDS-PAGE analyses of the gastroduodenal digests were performed as described by Benedeé et al. [16].

### 2.4. RP-HPLC

RP-HPLC analysis was performed using a Waters 600 HPLC instrument (Waters, Milford, MA, USA) with an RP318 column (250 × 4.6 mm, 5 µm of particle size, 300 A pore size, Bio-Rad). Samples were injected (50 µL) at 4 mg/mL, and eluted with 0.37% (*v*/*v*) trifluoroacetic acid (Scharlau Chemie, Barcelona, Spain) as solvent A, and 0.27% (*v*/*v*) trifluoroacetic acid in HPLC-grade acetonitrile (Lab-Scan, Gliwice, Poland) as solvent B. The elution of peptides was carried out using a linear gradient of solvent B in A ranging from 0% to 60% for 60 min at 1 mL/min. Detection was conducted at 220 nm and data were analysed by using Empower 2 Software from Waters.

### 2.5. Caco-2 Cell Culture

Human colon adenocarcinoma Caco-2 cells (ATCC, Rockville, MD, USA) were seeded at 2.5 × 10^5^ cells per well onto Transwell polycarbonate membrane supports (Costar, Corning, NY, USA) and cultured for 18–22 days. Fractions of the digests smaller than 10 kDa were added to the apical side of the plates (200 µg/mL) and incubated for 24 h at 37 °C and 5% CO_2_. The absence of lipopolysaccharides (LPSs) in the digests was verified in the transfected cell line THP1-XBlue™ using the QUANTI-Blue™ assay (Invitrogen, Carlsbad, CA, USA), following the manufacturer’s instructions. Caco-2 cells were conserved for gene expression analyses in buffer RLT (Qiagen, Hilden, Germany) and the supernatants were kept frozen at −80 °C until analysis for the detection of cytokine secretion by ELISA.

### 2.6. Cell Viability Assay

To assess cytotoxicity, lactate dehydrogenase (LDH) production by cells was measured in supernatants of culture cells using the CyQUANT™ LDH Cytotoxicity Assay Kit (Invitrogen) following the manufacturer’s instructions.

### 2.7. Enzyme Immunoassays

Supernatants of cultured cells collected 24 h after the addition of stimuli were used to measure IL-6, IL-8, IL-33, IL-25, and TSLP via an enzyme-linked immunosorbent assay (ELISA), following the manufacturer’s instructions (Thermo Fisher Scientific, Waltham, MA, USA).

### 2.8. Gene Expression

Gene expression analyses of the Caco-2 cells collected 24 h after the addition of stimuli were performed as described by Benedeé et al. [17]. The specific primers are described in Appendix A.

### 2.9. Statistical Analyses

Statistical analyses of the gene expression results were carried out using GraphPad Prism v6 (GraphPad Software Inc., San Diego, CA, USA). A one-way analysis of variance (ANOVA) was carried out for calculating statistical significance (*p* < 0.05). Results are presented as means ± SEM of 3 independent experiments.

## 3. Results

### 3.1. Impact of Heat Treatment on Digestibility of Egg White Proteins

The chromatographic profile of native and heated OVA, OM, and LYS and their respective in vitro gastric and duodenal digests are shown in Figure 2 and Appendix A. A noteworthy amount of intact OVA and LYS persisted after the digestion of native allergens, whereas OM was completely degraded. The heat treatment at 65 °C for 30 min did not alter the digestibility of OVA, whereas at 90 °C for 3 min, OVA degradation was favoured and no intact proteins were found at the end of digestion. The digestibility of OM and LYS was not affected by heat treatment. Chromatograms of the fractions with a molecular weight below 10 kDa of the gastroduodenal digests of OVA, OM, and LYS confirm the absence of intact proteins in these samples.

### 3.2. Influence of Peptides Released during Digestion of Egg White Proteins on Cytokine Production in Caco-2 Cells

In order to determine the impact of the peptides released during digestion on the production of proinflammatory cytokines in intestinal epithelial cells, Caco-2 cells were incubated for 24 h with the fractions of the digests containing peptides smaller than 10 kDa. The digests did not produce cytotoxicity as judged by the LDH production measured in the supernatants after incubation. Peptides obtained after the digestion of both native and heated OVA induced the production of the proinflammatory cytokines IL-6 and IL-8 (Figure 3), which was consistent with the expression of the genes that encode them (Figure 4). *Tslp* expression was also increased with the three samples of OVA peptides, although the heat treatment, irrespective of its intensity, applied to proteins before digestion reduced this effect, as did the gene expression of *Il6*. In the case of OM, all three digests promoted IL-6 and IL-8 release in a similar manner (Figure 3), as well as the gene expression of this cytokine, although peptides obtained after OM heating at 65 °C for 30 min did not affect the expression of *Il6* with respect to the control cells (Figure 4). Incubation of Caco-2 cells with the peptides smaller than 10 kDa from the three LYS digests resulted in the increased expression of *Il33* and *Il6* compared to control cells incubated with PBS alone, although no differences were observed between the different types of heat treatments that the proteins received prior to digestion (Figure 4). None of the samples studied induced the secretion of IL-33, IL-25, and TSLP or the expression of *Il25* (Figure 4).

## 4. Discussion

The heat treatment of food is often essential to guarantee microbial safety or to achieve pleasing organoleptic attributes. In the case of eggs, pasteurisation processes are applied to the whole food or their different parts separately (egg white and yolk) to extend shelf-life, but heat treatment is also used as a culinary technique in many food preparations and can modify their techno-unique features. Most studies focus on the microbiological quality and safety of the final egg-containing products or on the correlation between the structural characteristics of egg white proteins and their functional properties, although there are also a considerable number of studies that have evaluated the influence of heat treatments on egg protein digestibility because resistance to digestion could be used as a predictive factor in assessing the allergenic potential of a protein to induce an allergic reaction [9].

As shown by our results (Figure 2), previous studies have demonstrated that OVA and LYS can resist gastrointestinal digestion [16,18,19,20,21]. In contrast, OM was degraded during the simulated digestion, as previously described [20,22]. Intense heat treatments facilitated the degradation of OVA, as demonstrated in previous studies [20,23]. However, no differences in digestibility were found between native OVA and OVA heated at 65 °C for 30 min in the study by Jiménez-Sáiz et al. [20]; despite this, we observed a slight increase in the resistance to degradation, probably due to the different type of enzymes used in the duodenal phase of the digestion. These authors also described that a heat treatment at 95 °C for 15 min did not alter the digestibility of OM [20]. This difference could be due to the fact that the heat treatments used in this study are more moderate and the enzymes used for duodenal digestion are different.

Intestinal epithelial cells form a physical barrier against external antigens and constitute the interface of innate and adaptive immune responses [24]. In reaction to stress or cell injury, intestinal epithelial cells release the alarmin cytokines IL-33, IL-25, and TSLP. These cytokines are essential in the regulation of polarisation toward a Th2-type response, which is characteristic of the allergic condition, and play a leading role in the induction of allergic responses in the intestinal mucosa [25]. The tumour enterocyte cell line of human origin, Caco-2, has been one of the most predominantly used as a model to study interactions between food proteins and the epithelium [26], although most studies focus on assessing the disruption of tight junctions and allergen transport, and very few on the study of the regulatory role that the epithelium may play in the immune system. We observed that peptides obtained after the digestion of both native and heated OVA and OM induced the secretion of the proinflammatory cytokines IL-6 and IL-8. We did not observe an increase in the levels of these cytokines in the supernatants of Caco-2 cells treated with the peptides derived from the digestion of LYS, but we detected an increase in the expression of the gene encoding *Il6*. Although high levels of IL-6 have been found in patients with food allergies [27] and it has been shown that IL-6 can promote intestinal barrier permeability [28] and the suppression of regulatory T-cell generation [29], IL-8 promotes the recruitment of neutrophils and basophils to inflammation sites and is produced by IgE-binding monocytes in animals with naturally occurring allergies [30].

The epithelial cell-derived cytokines IL-33, IL-25, and TSLP are central regulators of type 2 immunity, which promotes a wide array of allergic responses, including food allergies [25]. Although we did not detect these cytokines in the supernatants of the analysed Caco-2 cells, we observed that peptides derived from OVA and LYS digestion promote the expression of *Tslp* and *Il33*, respectively. IL-33 plays an important role in food allergies since it has been shown that blocking antibodies against IL-33 prevented oral sensitisation to egg whites [31], and an anti-IL-33 antibody has shown promising results for the desensitisation of peanut-allergic individuals in a phase 2 clinical trial [32]. Khodoun et al. also observed that TSLP inhibition, as well as that of IL-25, also prevented sensitisation to egg proteins [31]. In addition, TSLP can promote lymphocyte proliferation and differentiation through the priming of dendritic cells toward an inflammatory Th2 phenotype [33].

The treatment prior to OVA digestion reduced IL-6 production and Tslp expression and, therefore, may exert a protective effect against egg sensitisation. Other studies have also shown that heat treatments could reduce the allergenicity and immunogenicity of egg proteins. Peptides released after digestion of heat-treated OVA and OM showed a significant reduction in their capacity to activate human basophils and degranulate RBL-2H3 cells [23]. A reduction in human IgG and the IgE binding of heat-treated digested OVA compared to the untreated digested protein has also been described [20]. In addition, extensive protein denaturation caused by heating at 80 °C for 10 min reduced both the sensitizing and eliciting capacity of egg white proteins [34]. However, despite these findings, we observed that a heat treatment at 90 °C for 3 min increased the digestibility of OVA compared to a treatment at 65 °C for 30 min, and we did not detect differences in the cytokine secretion induced in Caco-2 cells by digests generated after the digestion of OVA subjected to both heat treatments, probably because the stimulation of the cells was carried out with a peptide fraction under 10 kDa.

In summary, the data obtained in this work indicate that the heat treatment of eggs can reduce the release of peptides from OVA, but not from OM and LYS, with proinflammatory activity during digestion.

## Figures and Tables

**Figure 1 foods-11-03566-f001:**
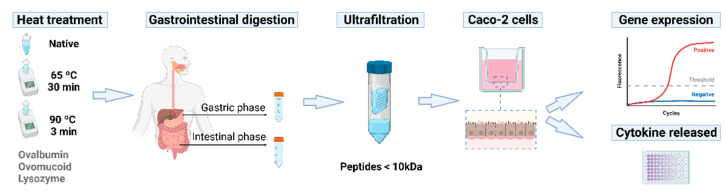
Schematic presentation of the experimental protocol used in this study for the determination of the cytokine production in Caco-2 cells after stimulation with peptides generated during gastrointestinal digestion of heated egg white proteins. Reproduced or adapted from [14], with permission from BioRender, 2022.

**Figure 2 foods-11-03566-f002:**
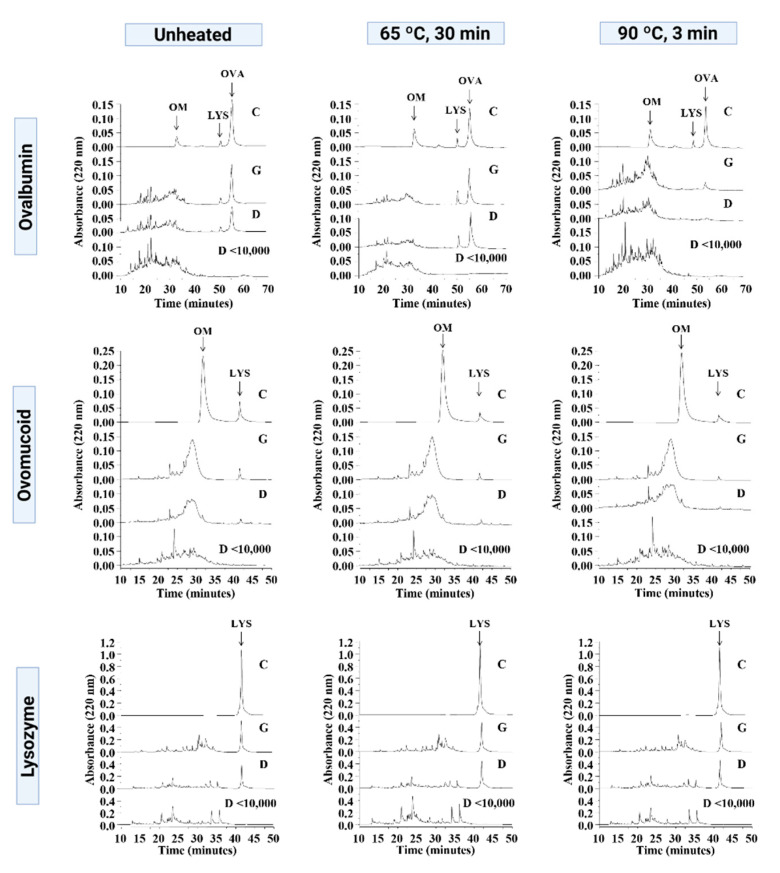
RP-HPLC chromatograms of undigested proteins (C), gastric digests (G), and gastroduodenal digests (D), including the fraction less than 10 kDa (D < 10,000), of ovalbumin (OVA), ovomucoid (OM), and lysozyme (LYS) subjected to two different heat treatments, 65 °C for 30 min and 90 °C for 3 min.

**Figure 3 foods-11-03566-f003:**
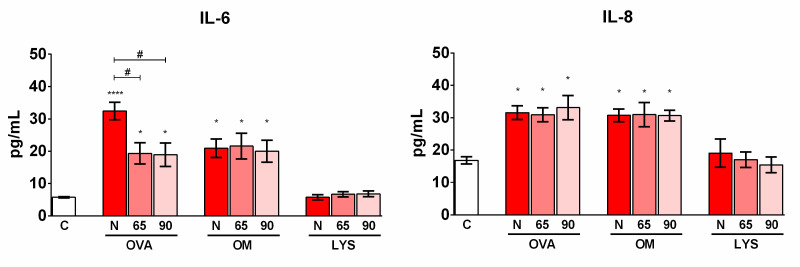
Secretion of IL-6 and IL-8 by Caco-2 cells after incubation with peptides (<10 kDa) generated during digestion of native (N) egg white proteins or subjected to two different heat treatments, 65 °C for 30 min (65) and 90 °C for 3 min (90). Asterisks and pounds indicate statistically significant differences with respect to the cells incubated with PBS (C) or between experimental groups, respectively. * and # *p* < 0.05; **** *p* < 0.0001.

**Figure 4 foods-11-03566-f004:**
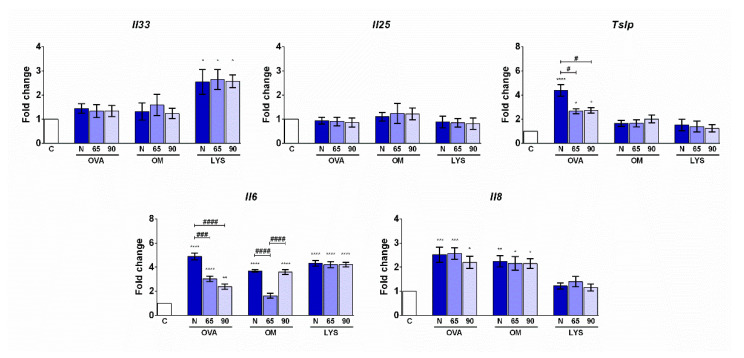
Relative gene expression of *Il33*, *Il25*, *Tslp*, *Il6*, and *Il8* in Caco-2 cells after incubation with peptides (<10 kDa) generated during digestion of native (N) egg white proteins or subjected to two different heat treatments, 65 °C for 30 min (65) and 90 °C for 3 min (90). Asterisks and pounds indicate statistically significant differences with respect to the cells incubated with PBS (C) or between experimental groups, respectively. * and # *p* < 0.05; ** *p* < 0.01; *** and ### *p* < 0.001; **** and #### *p* < 0.0001.

## Data Availability

Data is contained within the article or Appendix A.

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
