# Peer review of "Caco-2 Cell Response Induced by Peptides Released after Digestion of Heat-Treated Egg White Proteins"

_foods, 2022, doi:10.3390/foods11223566_

Round 1
Reviewer 1 Report
Authors present work on the effect of the peptides from egg proteins derived after simulated gastric digestion and using differently thermally treated proteins. The work has promising results however some modifications are required.
It would be useful to show an SDS page of digestion products of untreated and heat-treated allergens.
Why are results for ELISA detection of cytokines, except IL6 and IL8 missing?
What was used as a control for Caco2 stimulation?
Did you try to assess the effect of fragments larger than 10 kDa?
Did you try to asses allergenicity of the derived peptide in any other way?
Author Response
Comment 1: It would be useful to show an SDS page of digestion products of untreated and heat-treated allergens.
An SDS-PAGE of digestion products of untreated and heat-treated proteins has been included as supplementary figure.
Comment 2: Why are results for ELISA detection of cytokines, except IL6 and IL8 missing?
We measured the secretion of all cytokines (Il-33, IL-25. TSLP, IL-6, and IL-8) in the supernatants of Caco-2 cells but we did not detect IL-33, IL-25, and TSLP. This is indicated on lines 241 and 242 of the discussion and now we have also added a clarifying sentence on line 171 of the results section.
Comment 3: What was used as a control for Caco2 stimulation?
Caco-2 cells stimulated with PBS were used as a control.
Comment 4: Did you try to assess the effect of fragments larger than 10 kDa?
In this study we have only focused on the evaluation of peptides smaller than 10 KDa because we wanted to determine whether fragments with low IgE binding capacity due to their size, and which therefore a priori would not have much importance in the allergic response, could have any influence on the intestinal epithelium.
Comment 5: Did you try to asses allergenicity of the derived peptide in any other way?
We have not evaluated the allergenicity of peptides generated after digestion in this study, but we have published previously the IgE binding capacity of digested OVA, OM, and LYZ measured by ELISA and Western Blotting:
Benedé S, López-Expósito I, López-Fandiño R, Molina E. Identification of IgE-binding peptides in hen egg ovalbumin digested in vitro with human and simulated gastroduodenal fluids. J Agric Food Chem. 2014 ;62,1:152-8.
Benedé S, López-Fandiño R, Reche M, Molina E, López-Expósito I. Influence of the carbohydrate moieties on the immunoreactivity and digestibility of the egg allergen ovomucoid. PLoS One. 2013;8,11:e80810.
Jiménez-Saiz R, Benedé S, Miralles B, López-Expósito I, Molina E, López-Fandiño R. Immunological behavior of in vitro digested egg-white lysozyme. Mol Nutr Food Res. 2014;58,3:614-24.
Reviewer 2 Report
The manuscript titled "Caco-2 cell response induced by peptides generated after digestion of heat treated egg white proteins" is quite interesting. The experimental plan is well executed. However, it still needs major revisions.
1. The full name does not match the abbreviation, such as lysozyme (LYZ), LZ, LYS in the abstract.
2. In the method, the levels of IL-6, IL-8, IL-33, IL-25, and TSLP were measured by ELISA, but only IL-6 and IL-8 were shown. Please supplement the data of IL-33, IL-25, and TSLP.
3. The IL-6 release and gene expression of 65˚C heated OM are inconsistent, please analyze.
4. Why Il6 gene expression was significantly up-regulated in all LYS groups, but IL-6 release was not changed?
5. The results of this study showed that heat treatment significantly inhibited the expression of Il6 and Tslp genes in OVA-treated cells, but the authors did not measure the secretion of TSLP.
6. Tslp, Il6, and Il33 genes need to be represented in italics, such as line18.
7. Some content needs to be subscripted, such as line75, NaHCO3, line108, CO2.
8. Some content requires superscripts, such as Line79, U mg− 1, Line104, 2.5×105.
9. Line 267, Author Contributions is repetitive
10. The format of references is incorrect and inconsistent (references 2/21/24/30).
Author Response
Reviewer 2
The manuscript titled "Caco-2 cell response induced by peptides generated after digestion of heat treated egg white proteins" is quite interesting. The experimental plan is well executed. However, it still needs major revisions.
Comment 1: The full name does not match the abbreviation, such as lysozyme (LYZ), LZ, LYS in the abstract.
This has been corrected.
Comment 2: In the method, the levels of IL-6, IL-8, IL-33, IL-25, and TSLP were measured by ELISA, but only IL-6 and IL-8 were shown. Please supplement the data of IL-33, IL-25, and TSLP.
We measured the secretion of all cytokines (Il-33, IL-25. TSLP, IL-6, and IL-8) in the supernatants of Caco-2 cells but we did not detect IL-33, IL-25, and TSLP. This is indicated on lines 241 and 242 of the discussion and now we have also added a clarifying sentence on line 171 of the results section.
Comment 3: The IL-6 release and gene expression of 65˚C heated OM are inconsistent, please analyze.
Gene expression does not always match with cytokine released (Payne SH. The utility of protein and mRNA correlation. Trends Biochem Sci. 2015;40,1:1-3). This can be due to transcriptional regulation or post‐translational modifications, differences in protein in vivo half-lives, and in our case also because the supernatant and the cells were recovered at the same time point.
Comment 4: Why Il6 gene expression was significantly up-regulated in all LYS groups, but IL-6 release was not changed?
See reply to comment 3
Comment 5: The results of this study showed that heat treatment significantly inhibited the expression of Il6 and Tslp genes in OVA-treated cells, but the authors did not measure the secretion of TSLP.
We did not detect TSLP by ELISA in the supernatant of Caco-2 cells. This has been clarified in line 171 of the results section.
Comment 6: Tslp, Il6, and Il33 genes need to be represented in italics, such as line18.
This has been corrected.
Comment 7: Some content needs to be subscripted, such as line75, NaHCO3, line108, CO2.
This has been corrected.
Comment 8: Some content requires superscripts, such as Line79, U mg− 1, Line104, 2.5×105.
This has been corrected.
Comment 9: Line 267, Author Contributions is repetitive
This has been corrected.
Comment 10: The format of references is incorrect and inconsistent (references 2/21/24/30).
This has been corrected.
Round 2
Reviewer 2 Report
The authors addressed the concerns I raised.
Author Response
Dear reviewer,
Thank you for your time reviewing the manuscript and for your suggestions that have helped to improve it.
Kind regards
Sara